# Secretory leukocyte protease inhibitor influences periarticular joint inflammation in *Borrelia burgdorferi*-infected mice

Qian Yu[1]*[†], Xiaotian Tang[1,2][†], Thomas Hart[1], Robert Homer[3], Alexia A Belperron[4], Linda K Bockenstedt[4], Aaron Ring[5,6], Akira Nakamura[7], Erol Fikrig[1]

[1]Section of Infectious Diseases, Department of Internal Medicine, School of Medicine, Yale University, New Haven, United States; [2]Institute of Insect Sciences, College of Agriculture and Biotechnology, Zhejiang University, Hangzhou, China; [3]Department of Pathology, Yale School of Medicine, New Haven, United States; [4]Section of Rheumatology, Allergy and Immunology, Department of Internal Medicine, School of Medicine, Yale University, New Haven, United States; [5]Department of Immunobiology, Yale School of Medicine, New Haven, United States; [6]Department of Pharmacology, Yale School of Medicine, New Haven, United States; [7]Divisions of Immunology, Faculty of Medicine, Tohoku Medical and Pharmaceutical University, Sendai, Japan

*For correspondence: qian.yu@yale.edu

[†]These authors contributed equally to this work

## eLife Assessment

This study presents a **valuable** finding on the role of secretory leukocyte protease inhibitors (SLPIs) in developing Lyme disease in mice infected with *Borrelia burgdorferi*. The evidence supporting the authors' claims is **solid**. However, several concerns raised by the reviewers remain unaddressed. This paper would be of interest to scientists in the infectious inflammatory disease field.

**Abstract** Lyme disease, caused by *Borrelia burgdorferi*, is the most common tick-borne infection in the United States. Arthritis is a major clinical manifestation of infection, and synovial tissue damage has been attributed to the excessive pro-inflammatory responses. The secretory leukocyte protease inhibitor (SLPI) promotes tissue repair and exerts anti-inflammatory effects. The role of SLPI in the development of Lyme arthritis in C57BL/6 mice, which can be infected with *B. burgdorferi* but only develop mild joint inflammation, was therefore examined. *Slpi*-deficient C57BL/6 mice challenged with *B. burgdorferi* had a higher infection load in the tibiotarsal joints and marked periarticular swelling compared to infected wild-type control mice. The ankle joint tissues of *B. burgdorferi*-infected *Slpi*-deficient mice contained significantly higher percentages of infiltrating neutrophils and macrophages. *B. burgdorferi*-infected *Slpi*-deficient mice also exhibited elevated serum levels of IL-6, neutrophil elastase, and MMP-8. Moreover, using a recently developed BASEHIT (**BA**cterial **S**election to **E**lucidate **H**ost-microbe **I**nteractions in high **T**hroughput) library, we found that SLPI directly interacts with *B. burgdorferi*. These data demonstrate the importance of SLPI in suppressing periarticular joint inflammation in Lyme disease.

## Introduction

Lyme disease is the most common tick-borne illness in the United States, affecting an estimated 500,000 people each year (*Kugeler et al., 2021*). The spirochete *Borrelia burgdorferi* is the causative

agent of Lyme disease and is primarily transmitted by *Ixodes scapularis* ticks in North America (*Mead, 2022*). Early administration of antibiotics is usually successful in the treatment of Lyme disease. However, between 2008 and 2015, arthritis was the major manifestation in a third of Lyme disease cases reported to the CDC (*Arvikar and Steere, 2022*; *Schwartz et al., 2017*). Musculoskeletal symptoms occur at all stages of Lyme disease, with migratory arthralgias in the early stages and frank arthritis occurring months later. Lyme arthritis can present as acute or intermittent self-resolving episodes or persistent joint swelling and pain, which, if left untreated, can lead to irreversible joint dysfunction and debilitation (*Arvikar and Steere, 2022*; *Steere et al., 1987*; *Miller and Aucott, 2021*). Although Lyme arthritis resolves completely with antibiotic therapy in most patients, a small percentage of individuals experience persistent joint inflammation for months or several years, termed post-infectious Lyme arthritis (*Arvikar and Steere, 2022*; *Steere et al., 1987*; *Lochhead et al., 2021*).

Studies of synovial fluid from Lyme arthritis patients found infiltrating polymorphonuclear cells (PMNs), IFN-γ-producing mononuclear cells, and large amounts of NF-κB-induced pro-inflammatory cytokines and chemokines, such as IL-6, CXCL10, and TNF-α (*Miller and Aucott, 2021*; *Shin et al., 2007*; *Lochhead et al., 2019b*; *Gross et al., 1998*). An inverse correlation between the robust IFN-γ signature and tissue repair has been demonstrated in the synovial tissue and fluid from patients with post-infectious Lyme arthritis (*Lochhead et al., 2019a*). This suggests that the dysregulated excessive pro-inflammatory responses inhibit tissue repair and lead to extensive tissue damage.

*B. burgdorferi* infection of laboratory mice causes an acute arthritis, the severity of which is mouse strain dependent (*Barthold et al., 1990*). *B. burgdorferi*-infected-C3H/HeN mice develop pronounced neutrophilic infiltration of periarticular structures and the synovial lining, which peaks in severity several weeks after infection (*Barthold et al., 1993*). In contrast, infection of *B. burgdorferi* C57BL/6 mice causes mild, if any, arthritis (*Ma et al., 1998*). On a C57BL/6 background, the immune-deficient *Rag1-/-* and SCID mice are also resistant to *B. burgdorferi*-induced arthritis, indicating that responses independent of humoral and cellular immunity contribute to the milder phenotype of disease in these animals (*Brown and Reiner, 1999*). Similar to Lyme arthritis in humans, neutrophils, macrophages, and signaling involving IFN-γ and NF-κB contribute to the severity of murine joint inflammation (*Ritzman et al., 2010*; *Miller et al., 2008*; *Brown et al., 2003*; *Anguita et al., 2002*).

The secretory leukocyte protease inhibitor (SLPI) is a 12 kDa, secreted, non-glycosylated, cysteine-rich protein (*Doumas et al., 2005*). It strongly inhibits serine proteases, especially neutrophil-derived serine proteases (NSPs), including cathepsin G (CTSG) and elastase (NE) (*Thompson and Ohlsson, 1986*; *Moreau et al., 2008*). It is secreted by epithelial cells at various mucosal surfaces and is also produced by neutrophils, macrophages, mast cells, and fibroblasts (*Maruyama et al., 1994*; *Sallenave et al., 1994*). The major function of SLPI is to inhibit excessive protease activity at sites of inflammation, thus promoting tissue repair and wound healing (*Ashcroft et al., 2000*; *Zhu et al., 2002*). SLPI also exerts anti-inflammatory function by inhibiting NF-κB activation in macrophages (*Taggart et al., 2005*; *Taggart et al., 2002*; *Jin et al., 1997*). The roles of neutrophils and NSPs have been extensively studied in rheumatoid arthritis, a condition sharing some similarities with Lyme arthritis (*Huet et al., 1992*; *Wilkinson et al., 2019*). NE and CTSG induce potent destruction of cartilage proteoglycan in vitro and in vivo, which contributes to rheumatoid arthritis progression (*McDonnell et al., 1993*). Some studies also demonstrated that SLPI inhibits joint inflammation and bone destruction (*Lee et al., 2016*; *Song et al., 1999*). However, the importance of SLPI and NSPs has not been studied in the context of Lyme disease.

Thus, in this study, we examined the role of SLPI in the development of murine Lyme arthritis caused by *B. burgdorferi*. Using the *Slpi*-deficient C57BL/6 mice, we observed a significant increase in the infection burden and marked periarticular swelling in the ankle joints compared to WT mice following *B. burgdorferi* infection. Significant increases in infiltrating neutrophils and macrophages were observed in the ankle joints of infected *Slpi*-deficient mice. Elevated serum levels of IL-6, neutrophil elastase, and MMP-8 in the infected *Slpi*-deficient mice were also observed, which can lead to the recruitment of neutrophils and macrophages exacerbating the periarticular swelling. We further demonstrated the direct interaction between SLPI and *B. burgdorferi*. This is the first study showing the importance of anti-protease–protease balance in the development of murine Lyme arthritis.

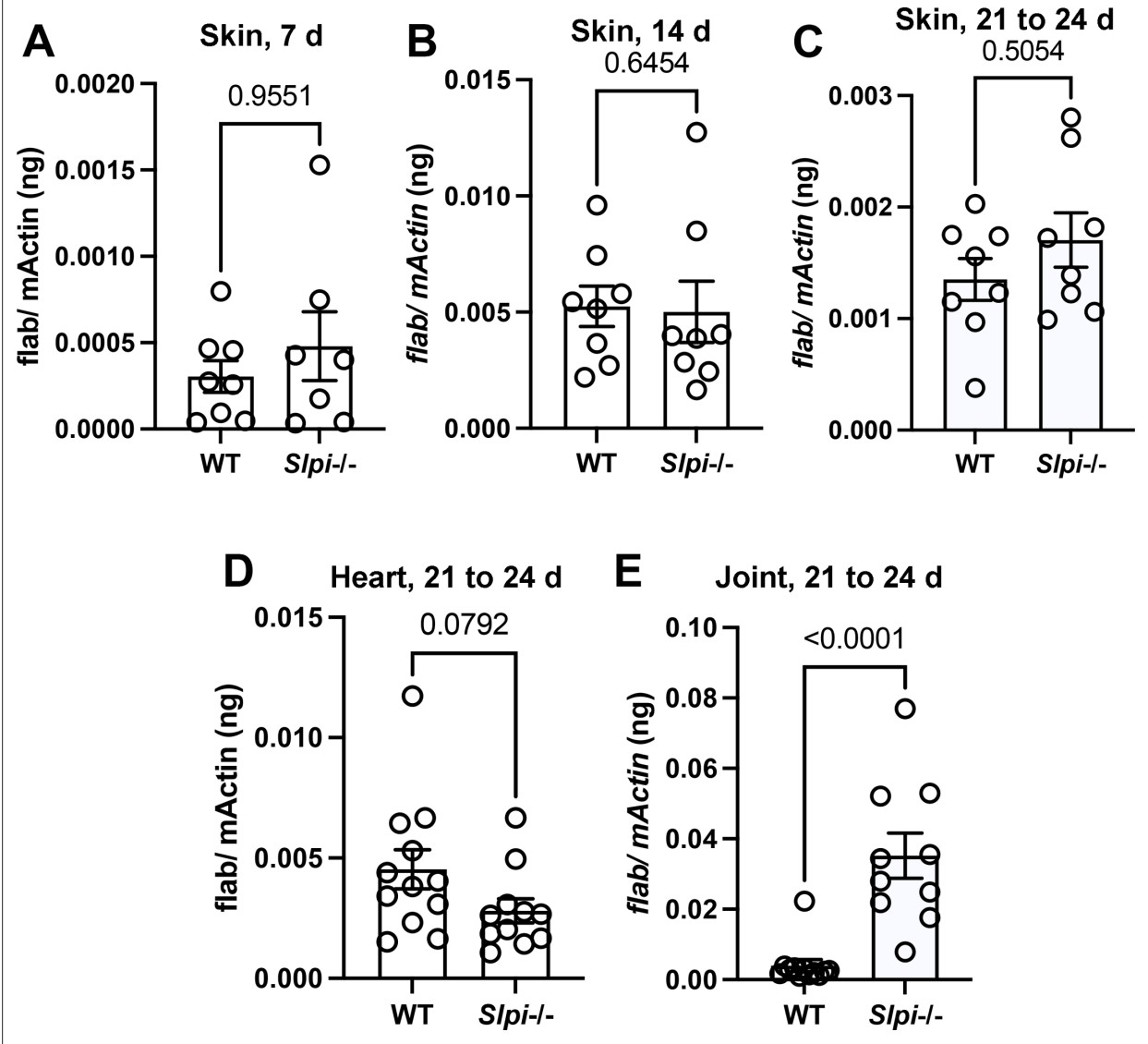

**Figure 1.** *B. burgdorferi* burden in C57BL/6 WT and *Slpi-/-* mice. WT and *Slpi-/-* mice were infected with 10$^5$ spirochetes by subcutaneous injection. (**A–C**) Spirochete burden in skin was assessed by ear punch biopsies at 7 days (**A**), 14 days (**B**), and between 21 and 24 days (**C**) post infection. (**D, E**) Spirochete burden in tibiotarsal joint and heart tissues was assessed between 21 and 24 days (**D**, heart, **E**, joint) post infection. At least n=6 mice were infected in each group. The spirochete burden was measured by qPCR detecting *flaB* and normalized to mouse *β-actin*. Each data point represents an individual animal. Representative data are shown from three separate experiments. The error bars represent mean ± SEM, and p-values were calculated using the nonparametric Mann–Whitney test.

The online version of this article includes the following source data for figure 1:

**Source data 1.** Source data value for *Figure 1A-E*.

## Results

### SLPI influences periarticular joint inflammation in *B. burgdorferi-* infected mice

To assess the importance of SLPI during murine Lyme arthritis, we compared the outcomes of *B. burgdorfe*ri infection of C57BL/6 WT and *Slpi-/-* mice. The C57BL/6 WT and *Slpi-/-* mice were infected with 10$^5$ spirochetes subcutaneously. Infection burdens in the skin were assessed by qPCR of *B. burgdorferi* DNA in ear punch biopsies at 7, 14, and between 21–24-day post-infection (dpi) (*Figure 1A–C*). Infection burdens in the heart (*Figure 1D*) and tibiotarsal joint (*Figure 1E*) tissues were assessed between

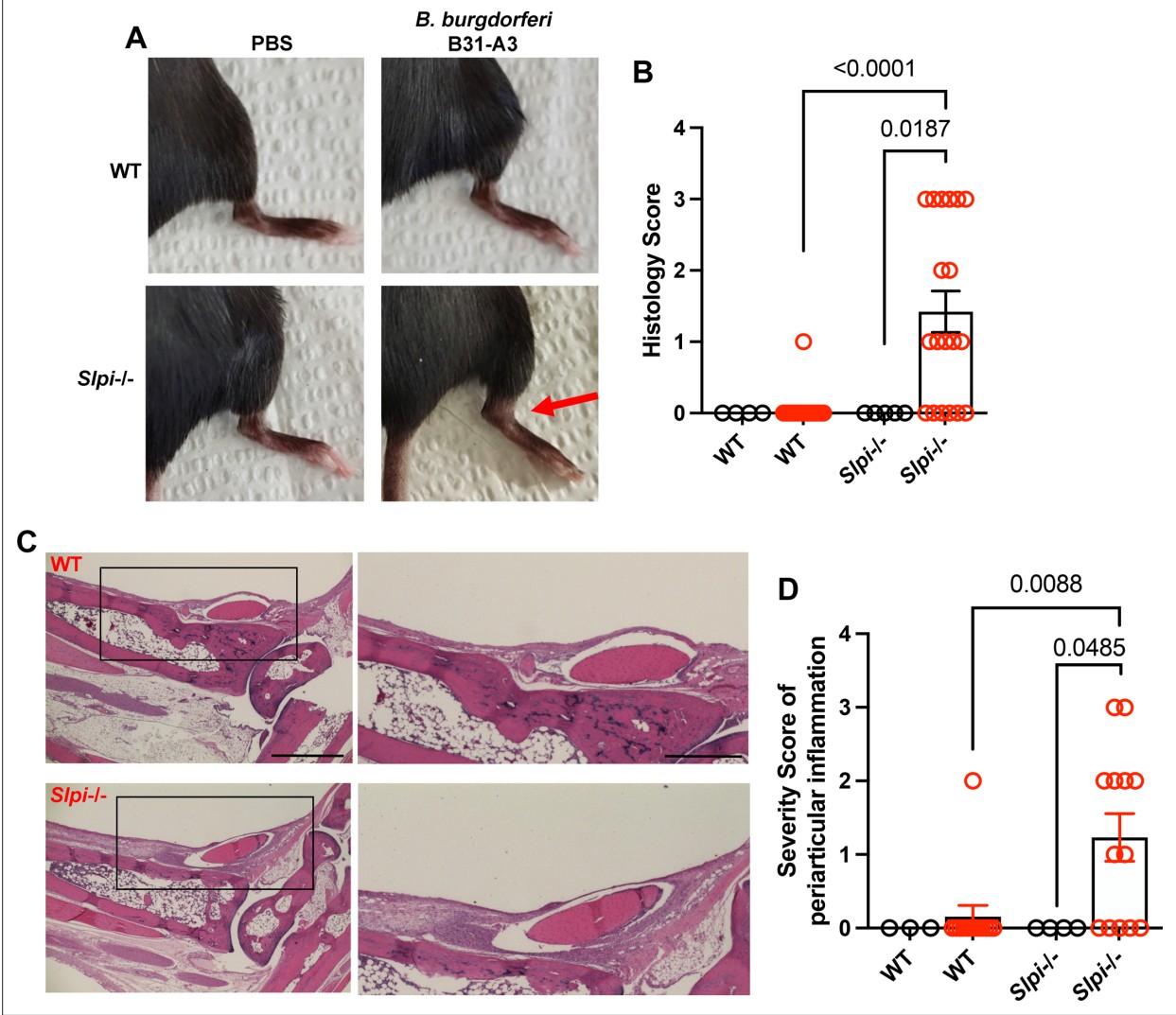

**Figure 2.** Assessment of ankle inflammation in WT and *Slpi-/-* mice infected with *B. burgdorferi* between 21 and 24 dpi. (**A**) Representative images are shown of the tibiotarsal joints of WT and *Slpi-/-* mice with/without *B. burgdorferi* infection between 21 and 24 dpi. The swelling is indicated by the red arrow. (**B**) Swelling of the tibiotarsal joints of individual mice was scored visually by an observer blinded to the experimental groups (scale of 0 [negative] to 3 [severe]). (**C**) The tibiotarsal joint of each mouse was dissected, fixed, sectioned, and stained with H&E. Representative images from *B. burgdorferi*-infected C57BL/6 WT and *Slpi-/-* mice are shown. Lower magnification (left panels, scale bar: 100 μm) and higher magnification (right panels, scale bar: 50 μm) of selected areas (black rectangle) are shown. (**D**) The severity of periarticular inflammation was scored blindly by the pathologist on a scale of 0 (negative) to 3 (severe). black, PBS-sham infection; red, *B. burgdorferi* infection. Results from two independent experiments were pooled and shown here. The error bars represent mean ± SEM, and p-values were calculated using the nonparametric Mann–Whitney test.

The online version of this article includes the following source data for figure 2:

**Source data 1.** Source data value for *Figure 2B and D*.

21 and 24 dpi. We did not observe any significant difference in infection burden in the skin between WT and *Slpi-/-* mice (n=24) at 7, 14, 21–24 dpi, or in the heart between 21–24 dpi (*Figure 1A–D*).

Strikingly, we observed a significantly higher spirochete burden in the ankle joints of infected *Slpi-/-* mice (n=24) between 21 and 24 dpi (*Figure 1E*). Furthermore, at around 24 dpi, significant swelling was also observed solely in the infected *Slpi-/-* mice (*Figure 2A*, red arrow). The level of swelling was first scored visually. While 14 out of 20 infected *Slpi-/-* mice displayed visible swelling (score ≥1), only 1 in 14 infected WT mice displayed mild swelling (score 1) at the ankle (*Figure 2B*). The tibiotarsal joints were then dissected, fixed, and stained with H&E for histopathological evaluation of the level of inflammation (*Figure 2C and D*). Inflammation of bursa and soft tissue adjacent to the tibiotarsal joint, but not in the tibiotarsal synovium, was consistently observed in the infected

*Slpi-/-* mice (*Figure 2C*, black rectangle). In contrast, only 1 out of 14 infected WT mice displayed modest inflammation (score = 2) in these sites (*Figure 2D*). The above data indicate the importance of SLPI in modulating the development of periarticular inflammation associated with murine Lyme arthritis.

## SLPI influences immune responses in *B. burgdorferi*-infected mice

It has been established that SLPI exerts its anti-inflammatory effect by inhibiting neutrophil serine protease (NSP) and dampening NF-κB activation in macrophages (*Moreau et al., 2008*). Thus, to investigate the mechanism underlying the effect of SLPI on murine joint inflammation, we sought to identify the population of infiltrating cells in the periarticular tissues of infected WT and *Slpi-/-* mice. Between 21 and 24 dpi, the ankle joints were dissected. To obtain single-cell suspensions of infiltrating cells, bone marrow cells were removed and discarded and the joint and periarticular tissues were digested (*Akitsu and Iwakura, 2016*). The cells were stained for flow cytometry with CD45, CD11b, and LY6G to label neutrophils (*Figure 3A*), and CX3CR1, CD64, and LY6C to label macrophages (*Figure 3B*; *Feng et al., 2023*). After gating, we observed significantly higher percentages of infiltrating neutrophils and macrophages in the dissected tissues from infected *Slpi-/-* than WT mice (*Figure 3A and B*). To further eliminate the possibility of neutrophilic contamination within the macrophage population, we also implemented a Ly6G-negative gating strategy. The result showed a consistently higher percentage of macrophages in the infected *Slpi-/-* mice (*Figure 3B*, *Figure 3—figure supplement 1*). Using RT-qPCR on the tibiotarsal tissues extracted from *B. burgdorferi*-infected *Slpi-/-* mice, we detected increased gene expression of neutrophil chemoattractant receptor C-X-C motif chemokine receptor 2 (*Cxcr2*), monocyte chemoattractant protein 1 (*Mcp-1*), and its receptor C-C motif chemokine receptor 2 (*Ccr2*) (*Figure 3C–E*).

Furthermore, the serum cytokine/chemokine profile was assessed from uninfected and *B. burgdorferi*-infected WT and *Slpi-/-* mice between 21 and 24 dpi. We observed a significant increase in IL-6 in infected *Slpi-/-* mice (*Figure 3F*). IL-6 recruits and stimulates neutrophils, leading to the secretion of neutrophil-derived serine proteases including neutrophil elastase (NE) and cathepsin G (CTSG) (*Srirangan and Choy, 2010*). The lack of serine protease inhibitors, such as SLPI, can cause excessive protease activity and subsequent tissue damage and inflammation (*Wilkinson et al., 2019*). Indeed, using ELISA, we observed a significantly higher level of NE solely in the serum of infected *Slpi-/-* mice (*Figure 3G*). An increased serum level of NE was also observed in the *B. burgdorferi*-infected, arthritis-susceptible C3H/HeN mice at 21 dpi (*Figure 3H*). These data suggest that, in the absence of SLPI, excessive serine protease activity can exacerbate murine Lyme arthritis.

A correlation between IL-6, macrophages, metalloproteinases (MMPs), and articular cartilage destruction has been observed in the synovial tissue of RA patients (*Murphy and Nagase, 2008*). Elevated levels of host matrix metalloproteinases (MMPs) have also been found in the synovial fluid of Lyme arthritis patients and can cause excessive tissue damage (*Hu et al., 2001*). Thus, a mouse MMP 5-Plex Discovery Assay was used to explore the serum levels of different MMPs. We observed a significant increase in the levels of MMP-8 solely in the infected *Slpi-/-* mice (*Figure 3I*). Taken together, our data suggest that SLPI suppresses inflammation in *B. burgdorferi*-infected mouse joint tissues by potentially inhibiting neutrophil and macrophage infiltration and subsequent protease-mediated tissue destruction.

## Decreased serum level of SLPI in Lyme disease patients

Despite numerous studies of serum, synovial fluid, and tissue from Lyme arthritis patients, the importance of anti-protease–protease balance in Lyme arthritis has not been investigated (*Lochhead et al., 2021*). Based on our data obtained from the *Slpi*-deficient mice, we assessed the serum SLPI level in Lyme disease patients (*Figure 4*). Due to the limited samples available from Lyme arthritis patients, we included samples from Lyme disease patients who presented with earlier manifestations of Lyme disease. The serum level of human SLPI assessed by ELISA showed a significant decrease in the SLPI level in Lyme disease patients comparing to healthy adult controls (*Figure 4*). Similar to our data from *B. burgdorferi*-infected mice, this result suggests a correlation between the lack of SLPI and humans exhibiting clinical manifestations of Lyme disease, including arthritis.

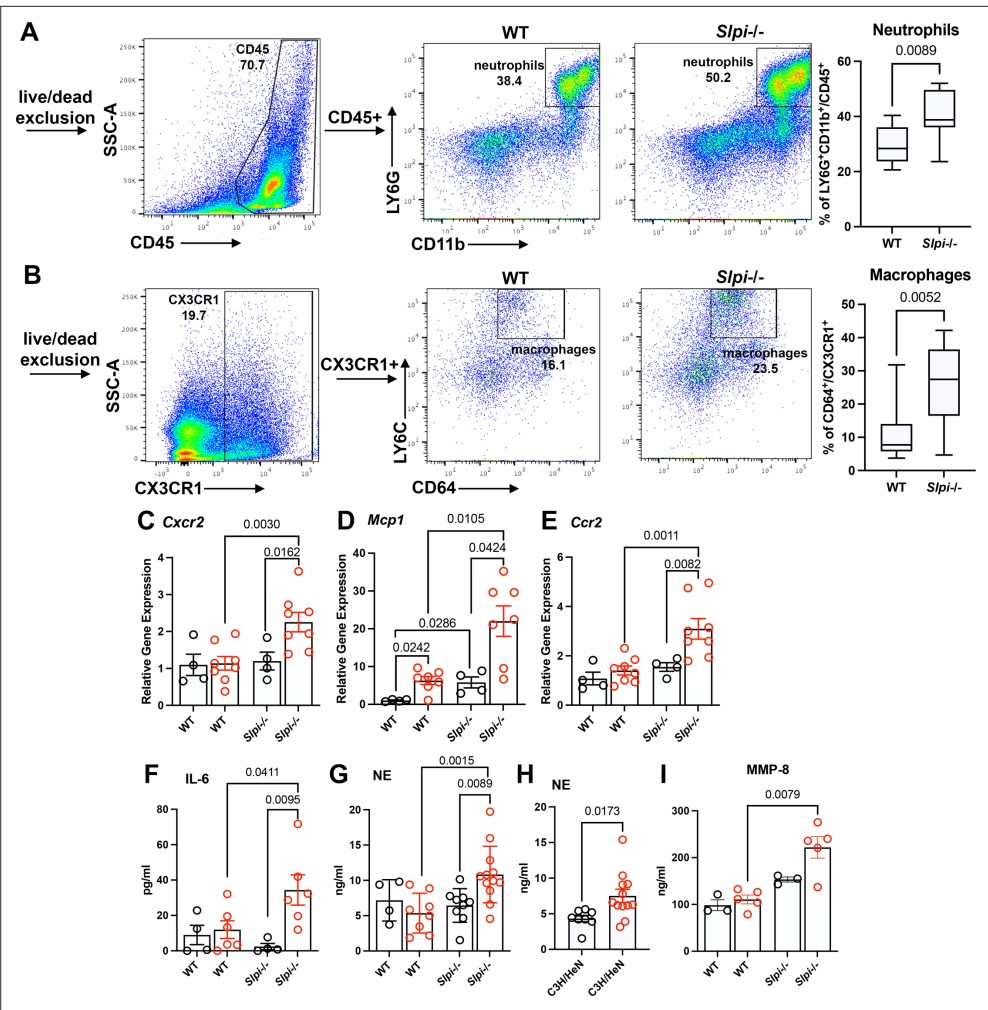

**Figure 3.** Immune profile analysis of infected WT and *Slpi-/-* mice. (**A, B**) Infiltrating cell population analysis of tibiotarsal joint tissues of infected WT and *Slpi-/-* mice. (**A**) The neutrophil population was gated on the CD11bLY6G double-positive cells among the CD45-positive cells. (**B**) The macrophage population was gated on the CD64-positive cells among the CX3CR1-positive myeloid cells. Results from two independent experiments were pooled and shown here. (**C–E**) Expression levels of C-X-C motif chemokine receptor 2 (*Cxcr2*, **C**), monocyte chemoattractant protein 1 (*Mcp-1*, **D**), and C-C motif chemokine receptor 2 (*Ccr2*, **E**) were assessed in the tibiotarsal tissue using RT-qPCR. (**F**) The serum cytokine profile was assessed using mouse cytokine/chemokine 32-plex array. An increase in IL-6 was observed in the infected *Slpi-/-* mice. (**G, H**) The serum level of neutrophil elastase (NE) was measured using an ELISA kit. (**I**) Serum levels of MMPs were assessed using a mouse MMP 5-Plex Discovery Assay. An increase in MMP-8 was observed in the infected *Slpi-/-* mice. Serum was obtained by cardiac puncture of WT and *Slpi-/-* C57BL/6 mice with/without infection between 21 and 24 dpi (**F**, **G**, and **I**) and of infected C3H/HeN mice at 21 dpi (**H**). black, PBS-sham infection; red, *B. burgdorferi* infection. Each data point represents an individual animal. The error bar represents mean ± SEM, and p-values were calculated using the nonparametric Mann–Whitney test.

The online version of this article includes the following source data and figure supplement(s) for figure 3:

**Source data 1.** Source data value for *Figure 3A-I*.

**Figure supplement 1.** The macrophages population analyzed using Ly6G-negative gating strategy.

**Figure supplement 1—source data 1.** Source data for *Figure 3-figure supplement 1*.

**Figure supplement 2.** Serum and gene expression levels of TNF-α.

**Figure supplement 2—source data 1.** Source data for *Figure 3-figure supplement 2*.

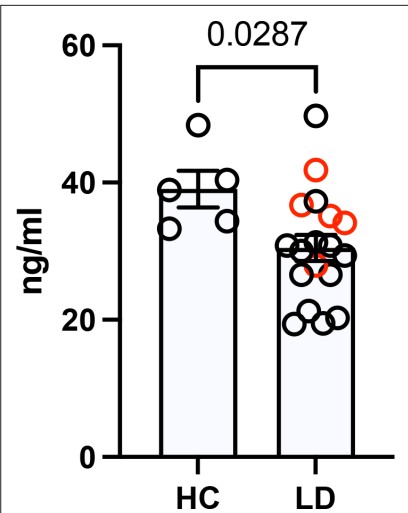

**Figure 4.** Serum secretory leukocyte protease inhibitor (SLPI) levels in Lyme disease subjects versus healthy controls. The serum level of SLPI was measured by ELISA. Sera samples were from five adult healthy controls (HCs). 18 samples were from people with Lyme disease (LD) including 5 samples from three subjects presenting with Lyme arthritis (red) and 13 samples from four subjects with erythema migrans (black). The error bar represents mean ± SEM, and p-values were calculated using the nonparametric Mann–Whitney test.

The online version of this article includes the following source data for figure 4:

**Source data 1.** Source data for **Figure 4**.

## SLPI interacts with *B. burgdorferi*

It has been demonstrated that *B. burgdorferi* interacts with various mammalian proteins to establish infection in the mammalian host (*Gupta et al., 2020*; *Shi et al., 2008*; *Li et al., 2006*; *Koenigs et al., 2013*). Thus, we postulated that *B. burgdorferi* could interact with SLPI to influence the progression of joint inflammation. To test this hypothesis, we probed a recently developed BASEHIT (**BA**cterial **S**election to **E**lucidate **H**ost-microbe **I**nteractions in high **T**hroughput) library with *B. burgdorferi* (*Gupta et al., 2020*; *Sonnert et al., 2024*; *Arora et al., 2022*; *Hart et al., 2024*). BASEHIT utilizes a genetically barcoded yeast display library expressing 3324 human exoproteins, thus enabling a comprehensive screen of host–microbe interactions in a high-throughput fashion. Human SLPI is one of the exoproteins that passed the significance threshold, indicating *B. burgdorferi*-SLPI binding. To further establish that hSLPI directly binds to *B. burgdorferi*, we performed ELISA with whole-cell *B. burgdorferi* lysates. We observed strong binding between whole-cell *B. burgdorferi* lysates and hSLPI at a level as low as 10 nM (*Figure 5A*).

To extend these studies, flow cytometry was performed using intact *B. burgdorferi* and both human and murine SLPI. A significant increase in fluorescence intensity was observed when *B. burgdorferi*, cultivated at 33°C, was incubated with human SLPI at 10 nM and 1 µM level (*Figure 5B*). Though the binding to 10 nM rmSLPI was at background level, we observed a significant increase in fluorescence intensity when *B. burgdorferi* were incubated with 1 µM rmSLPI (*Figure 5C*). Flow cytometry also demonstrated increased binding of *B. burgdorferi* cultured at 37°C to 1 µM hSLPI or rmSLPI (*Figure 5B and C*). This indicates that the binding was more robust when performed at temperatures that *B. burgdorferi* encounter in the mammalian host. Immunofluorescent microscopy was an additional method that also demonstrated direct binding of *B. burgdorferi* with human or murine SLPI (*Figure 5D*).

In contrast to *B. burgdorferi* B31-A3, an infectious strain used throughout this study, we did not observe any binding between hSLPI and *B. burgdorferi* B31A, a high-passage non-infectious strain (*Figure 5—figure supplement 1A*; *Caine and Coburn, 2015*; *Elias et al., 2002*). The above observation further suggests that the direct interaction between SLPI and *B. burgdorferi* could impact the pathogenesis of murine Lyme arthritis. To further investigate the potential *B. burgdorferi* ligand that interacts with SLPI, we probed protease-treated *B. burgdorferi* lysates with hSLPI using ELISA. After treatment with proteinase K, we observed a marked decrease in binding of hSLPI to *B. burgdorferi* lysates (*Figure 5—figure supplement 1B*). This result suggests that hSLPI can directly interact with a *B. burgdorferi* protein.

It has been showed that SLPI has antimicrobial effects against multiple gram-negative and -positive bacteria (*Wiedow et al., 1998*; *Nishimura et al., 2008*; *Hiemstra et al., 1996*). However, using the BacTiter-Glow microbial cell viability assay, we did not observe any significant changes in *B. burgdorferi* viability in the presence of hSLPI (*Figure 5—figure supplement 2A*). A previous study also demonstrated that the tick salivary protein, Salp15, specifically interacted with *B. burgdorferi* outer surface protein C (OspC) (*Ramamoorthi et al., 2005*). The binding of Salp15 protected spirochetes from killing by polyclonal mouse or rabbit antisera in vitro (*Ramamoorthi et al., 2005*; *Schuijt et al.,*

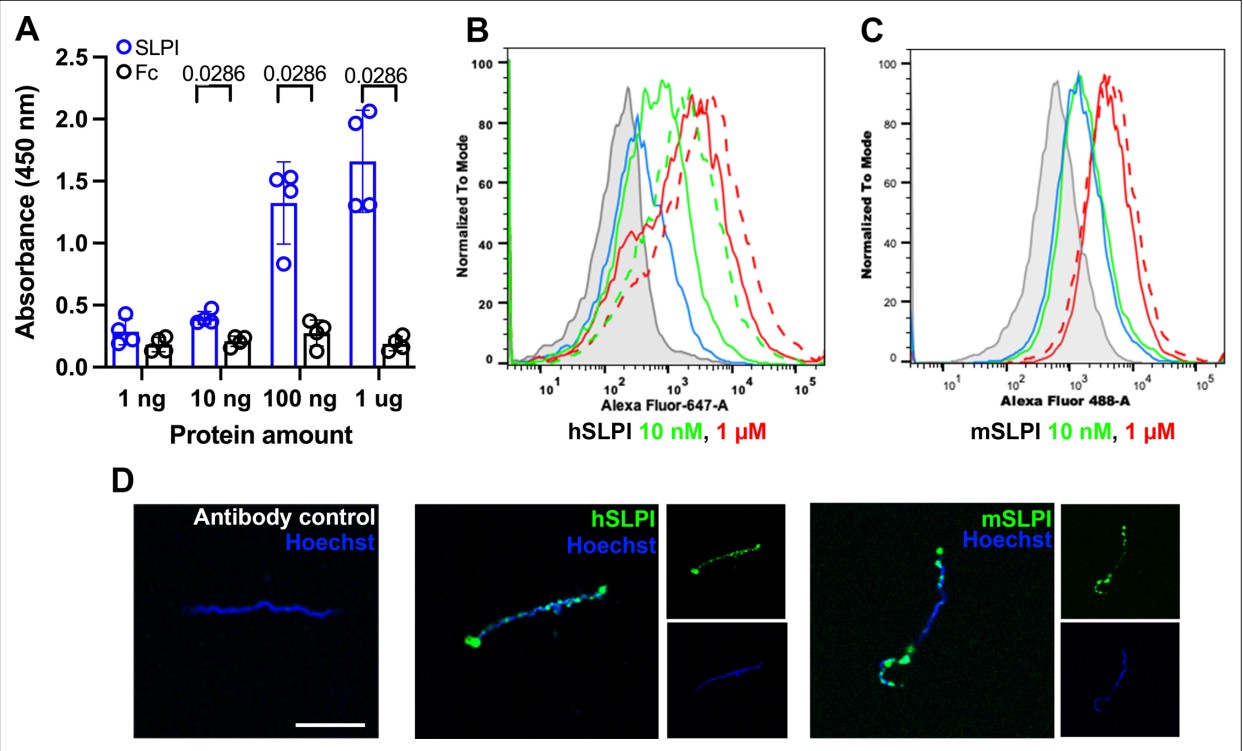

**Figure 5.** *B. burgdorferi* interaction with human and murine secretory leukocyte protease inhibitor (SLPI). (**A**) Sandwich ELISA results show the interaction of *B. burgdorferi* whole-cell lysates with human SLPI. ELISA plates were coated with *B. burgdorferi* whole-cell lysates and probed with increasing amount of human SLPI (blue) or human Fc proteins (black) as the negative control. The values plotted represent the mean ± SEM of duplicates from two experiments. p-value is displayed in the graph and determined using the nonparametric Mann–Whitney test. (**B, C**) Flow cytometry histograms show binding of human (**B**) and murine (**C**) SLPI to *B. burgdorferi* cultured at 33°C (solid line) and 37°C (dash line). *B. burgdorferi* was cultured to a density of $10^6$ /ml. The same volume of cultures was incubated at 33°C or 37°C for 24 h before adding 10 nM (green) or 1 μM (red) human or murine SLPI. The binding was detected with goat anti-human or murine SLPI and donkey anti-goat AF647 or AF488. *B. burgdorferi* alone (gray) and antibody control (without SLPI, blue) were used as negative controls. (**D**) Immunofluorescent microscopy was used to directly observe the binding of *B. burgdorferi* with human and murine SLPI. Merged and single-color images are shown. Representative histograms and fluorescent images are shown from three independent experiments. Scale bar: 10 μm.

The online version of this article includes the following source data and figure supplement(s) for figure 5:

**Source data 1.** Source data for *Figure 5A*.

**Figure supplement 1.** The binding of human secretory leukocyte protease inhibitor (SLPI) to non-infectious *B. burgdorferi B31A* and proteinase K-treated *B. burgdorferi B31A3*.

**Figure supplement 1—source data 1.** Source data for *Figure 5-figure supplement 1B*.

**Figure supplement 2.** The effect of human secretory leukocyte protease inhibitor (SLPI) binding on *B. burgdorferi* viability and antibody-mediated killing.

**Figure supplement 2—source data 1.** Source data for *Figure 5-figure supplement 2A and B*.

**Figure supplement 3.** Hoechst 33324 and propidium iodide double staining of *B. burgdorferi* whole organism with and without fixation.

*2008*). However, again, using the BacTiter-Glow microbial cell viability assay, the pre-incubation of hSLPI did not protect spirochetes from killing by mouse *B. burgdorferi* antisera (*Figure 5—figure supplement 2B*). Thus, the importance of the SLPI-*B. burgdorferi* interaction and the direct effect of such an interaction on *B. burgdorferi* biology are likely independent of direct borreliacidal activity or any interference with the antibody-mediated *B. burgdorferi* killing.

## Discussion

Lyme arthritis has been extensively documented and studied in patients and *B. burgdorferi*-infected mice. The pathogenesis of Lyme arthritis is characterized by synovial tissue damage caused by infiltration of immune cells and excessive pro-inflammatory responses (*Lochhead et al., 2021*). Transcriptomic

studies also revealed the suppression of tissue repair genes in the synovial tissue of Lyme arthritis patients and tibiotarsal joint tissues of *B. burgdorferi*-infected mice (*Lochhead et al., 2019a*; *Crandall et al., 2006*). However, the roles of the genes involved in tissue repair have not been studied.

SLPI strongly inhibits serine proteases, especially cathepsin G and elastase secreted by neutrophils (*Doumas et al., 2005*). The major function of SLPI is to prevent unnecessary tissue damage caused by excessive protease activity, thus promoting tissue repair and homeostasis (*Nugteren and Samsom, 2021*). The lack of SLPI impairs wound healing and tissue repair (*Ashcroft et al., 2000*), and SLPI also inhibits NF-κB activation and downstream pro-inflammatory cytokine release from macrophages (*Taggart et al., 2005*; *Jin et al., 1997*). Thus, we hypothesized that SLPI plays an important role in Lyme arthritis. To test this hypothesis, we employed *Slpi-/-* C57BL/6 mice. As C57BL/6 mice only develop mild arthritis, if any, after challenge with *B. burgdorferi*, this mouse provided an ideal example to study whether the lack of SLPI could cause an arthritis-resistant mouse to become arthritis-susceptible. Compared to WT C57BL/6 mice, *B. burgdorferi* infection in the *Slpi-/-* mice consistently showed a significantly higher infection burden in tissues extracted from the ankle joint, which included periarticular structures (*Figure 1E*). Severe swelling and inflammation in the bursa and soft tissue around tibiotarsal joints were observed solely in the infected *Slpi-/-* mice (*Figure 2*). The significant increase in the infection burden in *Slpi-/-* mice can contribute to the enhanced periarticular inflammation following *B. burgdorferi* infection. However, these data also suggest the importance of SLPI in controlling the development of inflammation in periarticular tissues of *B. burgdorferi*-infected mice. Indeed, in a *Streptococcal* cell wall (SCW)-induced arthritis model in rats, the intraperitoneal injection of SLPI significantly decreased the severity of joint swelling (*Song et al., 1999*). Targeting the SLPI-associated anti-protease pathways could also potentially be a strategy for ameliorating periarticular inflammation that occurs in some rheumatic diseases.

Analysis of *B. burgdorferi* infection in the *Slpi-/-* and WT mice revealed a significant increase in infiltrating neutrophils and macrophages in the periarticular tissues of *Slpi-/-* mice (*Figure 3A and B*). This observation is consistent with clinical studies that showed a high percentage of neutrophils in the synovial fluid during active infection (*Lochhead et al., 2017*; *Grillon et al., 2019*). In post-infectious Lyme arthritis, fewer neutrophils and more macrophages are present in patients' synovial fluid (*Lochhead et al., 2021*). In arthritis-susceptible C3H/He mice, *B. burgdorferi* infection also leads to neutrophil infiltration in the periarticular tissues as well as in the synovium of ankle joints (*Barthold et al., 1993*). The neutrophil chemoattractant KC (CXCL1) and receptor CXCR2 mediates neutrophil recruitment and is critical for the development of murine Lyme arthritis. Both *Kc-/-* and *Cxcr2-/-* C3H mice developed significantly less ankle swelling when infected with *B. burgdorferi* (*Ritzman et al., 2010*; *Brown et al., 2003*). Consistently, we observed a significant increase in *Cxcr2* gene expression in the tibiotarsal joint tissues (*Figure 3C*), which can recruit neutrophils and cause more severe inflammatory soft-tissue infiltrates in the *Slpi-/-* mice. Monocyte chemoattractant protein-1 (MCP-1/CCL2) and receptor CCR2 contribute to macrophage infiltration (*Deshmane et al., 2009*). Though little to no difference in arthritis was observed in the *Ccr2-/-* mice, a high level of MCP-1 was detected in the tibiotarsal tissues of *B. burgdorferi*-infected, arthritis-susceptible C3H/He mice, suggesting a function for macrophages (*Brown et al., 2003*). In the infected *Slpi-/-* mice, a significant increase in both *Mcp-1* and *Ccr2* gene expression was observed in the tibiotarsal tissues (*Figure 3D and E*). Our data suggest that both neutrophils and macrophages contribute to the severe periarticular inflammation in the *B. burgdorferi*-infected *Slpi-/-* mice.

To investigate the underlying mechanism whereby SLPI deficiency enhanced periarticular joint inflammation, we examined the serum cytokine/chemokine profile of *B. burgdorferi*-infected *Slpi-/-* and WT mice. There was a significant increase in IL-6 solely in the infected *Slpi-/-* mice (*Figure 3F*). An elevated IL-6 level has been demonstrated in the serum, synovial fluid, and synovial tissue from Lyme arthritis patients (*Lochhead et al., 2019b*; *Lochhead et al., 2017*). IL-6 is also pivotal in the pathogenesis of rheumatoid arthritis and correlates with the disease severity and joint destruction (*Srirangan and Choy, 2010*). It has been shown that IL-6 recruited neutrophils in an in vitro co-culture rheumatoid arthritis model (*Lally et al., 2005*). Neutrophils can be activated by IL-6 through binding of IL-6 receptor (IL-6R) (*Srirangan and Choy, 2010*). Activated neutrophils release several NSPs including neutrophil elastase (NE), cathepsin G (CTSG), and proteinase-3 (PR3), which can lead to potent cartilage destruction (*Wilkinson et al., 2019*). Indeed, we observed a significant increase in the NE level in the serum of *B. burgdorferi*-infected *Slpi-/-* but not WT mice (*Figure 3G*). In the arthritis-susceptible

C3H/HeN mice, *B. burgdorferi* infection also induced a significant increase in the serum NE level (*Figure 3H*). Interestingly, despite the known function of TNF-α in inflammatory responses, we did not observe any significant changes in either TNF-α serum protein levels or *Tnf-α* gene expression levels in the *B. burgdorferi*-infected *Slpi-/-* and WT mice (*Figure 3—figure supplement 2*). The result is consistent with previous microarray data that did not show significant changes in TNF-α levels in the C57BL/6 mice following *B. burgdorferi* infection (*Crandall et al., 2006*). The above data indicate that the excessive serum neutrophil elastase contributed to the periarticular inflammation in the *B. burgdorferi*-infected *Slpi-/-* mice.

MMPs target extracellular matrix and cause articular cartilage destruction (*Grillet et al., 2023*). A correlation between IL-6 and MMPs expression has been reported in the context of rheumatoid arthritis (*Murphy and Nagase, 2008*; *Chang et al., 2008*). Elevated levels of several MMPs have also been found in the synovial fluid of Lyme arthritis patients (*Hu et al., 2001*). Thus, we also investigated the MMPs profile in the *B. burgdorferi*-infected *Slpi-/-* and WT mice. We observed a significant increase in the serum level of MMP-8 in the infected *Slpi-/-* mice (*Figure 3I*). MMP-8 is known as neutrophil collagenase (*Wen et al., 2015*). Using synovial fluid samples, it has been reported that the level of MMP-8 was significantly higher in the patients with post-infectious Lyme arthritis than patients with active infection (*Lin et al., 2001*). A comprehensive examination of the MMP profile in the synovial fluid of Lyme arthritis patients revealed elevated levels of MMP-1, -3, -13, and -19 (*Behera et al., 2005*). *B. burgdorferi* infection induced MMP-3 and MMP-19 in the C3H/HeN mice but not in the Lyme arthritis-resistant C57BL/6 mice (*Behera et al., 2005*). The differences in the MMP profiles provide an explanation for the differences between human and murine Lyme arthritis. This finding further emphasizes that excessive protease activity can contribute to the severity of periarticular inflammation in *B. burgdorferi*-infected mice.

Previous research using serum, synovial fluid, and tissue from Lyme arthritis patients has been heavily focusing on innate and adaptive immune responses (*Lochhead et al., 2021*). As a result, limited data can be found regarding anti-protease and protease responses during Lyme arthritis in human patients. In this study, we tested serum SLPI level in five healthy subjects, eight Lyme disease patients, three of whom had overt arthritis. Though the number of healthy subjects is small, the median level of SLPI tested here (38.92 ng/ml, *Figure 4*) is comparable with previous studies showing in average about 40 ng/ml SLPI in the serum from healthy volunteers (*Zakrzewicz et al., 2019*; *Grobmyer et al., 2000*). While the clinical manifestation of five of the patients with Lyme disease was an EM skin lesion (*Supplementary file 1*), some symptoms persisted several months after diagnosis, a timeframe when acute arthritis often develops. We observed decreased SLPI in the serum of these patients (*Figure 4*), suggesting an inverse relationship between the SLPI level and *B. burgdorferi* infection. However, a large number of sera and synovial fluid samples from Lyme arthritis patients and other clinical manifestations of Lyme disease are needed to establish a definitive association.

*B. burgdorferi* first infects the skin of a vertebrate host following a tick bite, then disseminates throughout the body, colonizes various tissue, evades immune responses, and persists for a significant period of time. To survive the above processes, *B. burgdorferi* interacts with various mammalian proteins, including decorin (*Shi et al., 2008*), fibronectin (*Li et al., 2006*), and plasminogen (*Koenigs et al., 2013*), among others. To comprehensively study the potential interaction between *B. burgdorferi* and the host, our lab employed the BASEHIT to assess the interactions between *B. burgdorferi* and 3336 human extracellular and secreted proteins (*Sonnert et al., 2024*; *Hart et al., 2024*). Using BASEHIT, our lab previously identified a strong interaction between *B. burgdorferi* and Peptidoglycan Recognition Protein 1 (PGYRP1) (*Gupta et al., 2020*). Increased infection burden in the heart and joint was observed in the mice lacking PGYRP1, suggesting a role of PGYRP1 in the host response to *B. burgdorferi* infection. Expanding the use of BASEHIT, CD55 was identified to bind *Borrelia crocidurae* and *Borrelia persica*, two pathogens causing relapsing fever (*Arora et al., 2022*). CD55-deficient mice infected with *B. crocidurae* displayed lower pathogen load and elevated pro-inflammatory cytokines. The above data demonstrate BASEHIT as an effective method to identify host factors important in *B. burgdorferi* pathogenesis in vivo. In this study, we identified an interaction between SLPI and *B. burgdorferi* using BASEHIT library screening and subsequent flow cytometric analysis (*Figure 5*). The antimicrobial activity of SLPI has been demonstrated against both gram-negative and -positive bacteria, including *Escherichia coli*, *Pseudomonas aeruginosa* (*Wiedow et al., 1998*), *Mycobacteria tuberculosis* (*Nishimura et al., 2008*), *Staphylococcus aureus* (*Hiemstra et al., 1996*), and *Staphylococcus*

*epidermidis* (*Wiedow et al., 1998*). Interaction between the positive charges of SLPI and the negative charges of bacteria surface, including lipopolysaccharide (LPS), can destabilize bacteria cell wall leading to the bactericidal effect (*Tarhini et al., 2018*). *B. burgdorferi* do not have LPS (*Takayama et al., 1987*), and this may account for the absence of the bactericidal effect of SLPI against *B. burgdorferi* (*Figure 5—figure supplement 2A*). Future research is needed to understand the significance of the SLPI-*B. burgdorferi* binding in the development of periarticular inflammation. The potential *B. burgdorferi* protein that interact with SLPI remains unknown. It is our hypothesis that SLPI may bind and inhibit an unknown *B. burgdorferi* virulence factor that could contribute to the development of murine Lyme arthritis.

In conclusion, our data demonstrated the importance of SLPI in suppressing *B. burgdorferi*-induced periarticular inflammation in mice by inhibiting recruitment of neutrophils and macrophages and subsequent protease levels. We propose that, during the active infection of the murine joint structures, the binding of *B. burgdorferi* with SLPI depletes the local environment of SLPI. Such binding is specific to the infectious strain of *B. burgdorferi*. As a potent anti-protease, the decrease in SLPI results in excessive protease activity, including neutrophil elastase and MMP-8. These unchecked proteases can lead to extensive tissue inflammation. Our study is the first to emphasize the importance of an anti-protease–protease balance in the development of the periarticular inflammation seen in *B. burgdorferi*-infected mice.

# Materials and methods

**Key resources table**

| Reagent type (species) or resource | Designation | Source or reference | Identifiers | Additional information |
|---|---|---|---|---|
| Strain, strain background (*Borrelia burgdorferi*) | B31-A3 | Dr. Utpal Pal's laboratory | N/A | See 'Materials and methods', '*B. burgdorferi* culture' |
| Strain, strain background (*B. burgdorferi*) | B31-A | This paper | N/A | See 'Materials and methods', '*B. burgdorferi* culture' |
| Strain, strain background (*Escherichia coli*) | Rosetta-gami 2 (DE3) | Novagen | Cat#71351 | Electrocompetent cells |
| Strain, strain background (*Mus musculus*) | *SLPI-/-* | Dr. Akira Nakamura's laboratory (*Takayama et al., 1987*; *Bernard et al., 2018*) | N/A | https://doi.org/10.3389/fimmu.2017.01538 https://doi.org/10.1084/jem.20021824 |
| Strain, strain background (*M. musculus*) | C3H/HeN | Charles River Laboratories | N/A | |
| Strain, strain background (*M. musculus*) | C57BL/6 | Jackson Laboratory | Stock #: 000664 RRID:IMSR_JAX:000664 | |
| Biological samples (*M. musculus*) | Mouse tibiotarsal tissue | This paper | N/A | Freshly isolated from *Mus musculus* |
| Antibody | TruStain FcX anti-mouse CD16/32 | BioLegend | Cat#101320 RRID:AB_1574975 | Flow cytometry (1 µl per test) |
| Antibody | PerCP anti-mouse CD45 | BioLegend | Cat#103130 RRID:AB_893339 | Flow cytometry (1 µl per test) |
| Antibody | BV711 anti-mouse Ly6G | BioLegend | Cat#127643 RRID:AB_2565971 | Flow cytometry (1 µl per test) |
| Antibody | PE anti-mouse CD11b | BioLegend | Cat#101208 RRID:AB_312791 | Flow cytometry (1 µl per test) |

*Continued on next page*

*Continued*

| Reagent type (species) or resource | Designation | Source or reference | Identifiers | Additional information |
|---|---|---|---|---|
| Antibody | APC/CY7 anti-mouse CX3CR21 | BioLegend | Cat#149047 RRID:AB_2892303 | Flow cytometry (1 µl per test) |
| Antibody | FITC anti-mouse Ly6C | BioLegend | Cat#128005 RRID:AB_1186134 | Flow cytometry (1 µl per test) |
| Antibody | APC anti-mouse CD64 | BioLegend | Cat#139305 RRID:AB_11219205 | Flow cytometry (1 µl per test) |
| Antibody | Goat anti-human SLPI | R&D Systems | Cat#AF1274 RRID:AB_2302508 | Flow cytometry (1 µl per test) |
| Antibody | Goat anti-murine SLPI | R&D Systems | Cat#AF1735 RRID:AB_2195050 | Flow cytometry (1 µl per test) |
| Antibody | Alexa Fluor 488 donkey anti-goat IgG (H+L) | Invitrogen | Cat#A32814 RRID:AB_2762838 | Flow cytometry (1 µl per test) |
| Antibody | Alexa Fluor 647 donkey anti-goat IgG (H+L) | Invitrogen | Cat#A-21447 RRID:AB_2535864 | Flow cytometry (1 µl per test) |
| Recombinant DNA reagent | Murine SLPI cDNA ORF clone (plasmid) | GenScript | OMu22721 | |
| Recombinant DNA reagent | pET-22b (+) (plasmid) | Novagen | Cat#69744-3 | |
| Sequence-based reagent | Mouse *β-actin*-F | This paper | PCR primers | AGCGGGAAATCGTGCGTG |
| Sequence-based reagent | Mouse *β-actin*-R | This paper | PCR primers | CAGGGTACATGGTGGTGCC |
| Sequence-based reagent | *Borrelia flab*-F | This paper | PCR primers | TTCAATCAGGTAACGGCACA |
| Sequence-based reagent | *Borrelia flab*-R | This paper | PCR primers | GACGCRRGAGACCCTGAAAG |
| Sequence-based reagent | Mouse *Mcp1*-F | This paper | PCR primers | GTTGGCTCAGCCAGATGCA |
| Sequence-based reagent | Mouse *Mcp1*-R | This paper | PCR primers | AGCCTACTCATTGGGATCATCTTG |
| Sequence-based reagent | Mouse *Ccr2*-F | This paper | PCR primers | AGTAACTGTGTGGATTGACAAGCACTTAGA |
| Sequence-based reagent | Mouse *Ccr2*-R | This paper | PCR primers | CAACAAAGGCATAAATGACAGGAT |
| Sequence-based reagent | Mouse *Cxcr2*-F | This paper | PCR primers | CACCCTCTTTAAGGCCCACAT |
| Sequence-based reagent | Mouse *Cxcr2*-R | This paper | PCR primers | ACAAGGACGACAGCGAAGATG |
| Peptide, recombinant protein | human SLPI | R&D Systems | Cat#1274-PI-100 | |
| Commercial assay or kit | LIVE/DEADfixable violet stain kit | Invitrogen | Cat#L34955 | |
| Commercial assay or kit | DNeasy Blood & Tissue Kit | QIAGEN | Cat#69504 | |
| Commercial assay or kit | iScript cDNA Synthesis Kit | Bio-Rad | Cat#1708891 | |

*Continued on next page*

*Continued*

| Reagent type (species) or resource | Designation | Source or reference | Identifiers | Additional information |
|---|---|---|---|---|
| Commercial assay or kit | Gibson Assembly Kit | NEB | Cat#E5510 | |
| Commercial assay or kit | Mouse Neutrophil Elastase/ELA2 DuoSet ELISA | R&D Systems | Cat#DY4517-05 | |
| Commercial assay or kit | Human SLPI DuoSet ELISA | R&D Systems | Cat#DY1274-05 | |
| Commercial assay or kit | BacTiter-Glo Microbial Cell Viability Assay kit | Promega | Cat#G8230 | |
| Commercial assay or kit | Mouse MMP 5-Plex Discovery Assay Array (MDMMP-S, P) | Eve Technologies | N/A | |
| Commercial assay or kit | Mouse Cytokine/Chemokine 32-Plex Discovery Assay Array (MD32) | Eve Technologies | N/A | |
| Chemical compound, drug | iQ SYBR Green Supermix | Bio-Rad | Cat#1725124 | |
| Chemical compound, drug | Barbour-Stoenner-Kelly H (BSK-H) complete medium | Sigma-Aldrich | Cat#B8291 | |
| Chemical compound, drug | Bouin's solution | Sigma-Aldrich | Cat#HT10132 | |
| Chemical compound, drug | Hyaluronidase | Sigma-Aldrich | Cat#H3506 | |
| Chemical compound, drug | Collagenase | Sigma-Aldrich | Cat#C2139 | |
| Chemical compound, drug | ACK Lysing buffer | Gibco | Cat#A1049201 | |
| Chemical compound, drug | Trizol | Invitrogen | Cat#15596-018 | |
| Chemical compound, drug | Mca-RPKPVE-Nval-WRK(Dnp)-NH2 Fluorogenic MMP Substrate | R&D Systems | Cat#ES002 | |
| Chemical compound, drug | BugBuster Protein Extraction Reagent | Novagen | Cat#70921-3 | |
| Chemical compound, drug | Proteinase K | Thermo Scientific | Cat#EO0491 | |
| Chemical compound, drug | Ni-NTA agarose | QIAGEN | Cat#30230 | |
| Chemical compound, drug | KPL Sureblue TMB Microwell Peroxidase substrate, 1-component | Seracare | Cat#5120-0077 | |
| Chemical compound, drug | KPL TMB stop solution | Seracare | Cat#5150-0021 | |
| Chemical compound, drug | Hoechst 33342 | Invitrogen | Cat#H1399 | |
| Chemical compound, drug | RPMI 1640 | Gibco | Cat#11875-093 | |
| Software, algorithm | Prism | GraphPad | RRID:SCR_002798 | |

| Reagent type (species) or resource | Designation | Source or reference | Identifiers | Additional information |
|---|---|---|---|---|
| Software, algorithm | FlowJo | BD Biosciences | https://www.flowjo.com/ | |

## Sex as a biological variable

Females *Slpi-/-* and WT C57BL/6 mice were used for the in vivo *B. burgdorferi* infection. We have examined *B. burgdorferi* infection in both male and female C57BL/6 mice, and no differences in the development of infection or disease have been noted. Both male and female Lyme disease patients were included in the study. Sex was not considered a biological variable.

## Study approval

This study used archived serum samples from adult Lyme disease subjects and healthy controls that were previously collected under NIH U19AI089992 with approval of the Yale University Institutional Review Board for human subjects research (IRB protocol# 1112009475). All the animal experiments in this study were performed in accordance with the Guidelines for the Care and Use of Laboratory Animals of the National Institutes of Health. The animal protocol (2023-07941) was approved by the Institutional Animal Care and Use Committee at the Yale University School of Medicine.

## Measurement of serum SLPI levels in Lyme disease subjects and controls

SLPI levels were measured in a total of 23 serum samples from seven subjects at the time of Lyme disease diagnosis (four with a single erythema migrans lesion and three with the late manifestation of Lyme arthritis) and from five healthy controls. Serum samples from Lyme disease subjects were available at up to three times points: (1) study entry, range 0–9 days after onset of symptoms; (2) 30 days post diagnosis; and (3) up to 3 months after the completion of antibiotic therapy (range 4.5–6 months after diagnosis). Additional details can be found in *Supplementary file 1*. The level of SLPI in the serum was measured using the Human SLPI DuoSet ELISA kit (R&D Systems, #DY1274-05).

## *B. burgdorferi* culture

*B. burgdorferi* B31-A3, an infectious clonal derivative of the sequenced strain B31, was a generous gift from Dr. Utpal Pal at the Department of Veterinary Medicine, University of Maryland, College Park (*Bernard et al., 2018*). *B. burgdorferi* B31-A3 and *B. burgdorferi* B31A were grown in Barbour-Stoenner-Kelly H (BSK-H) complete medium (Sigma-Aldrich, #B8291) in a 33°C setting incubator. The live cell density was determined by dark field microscopy and using a hemocytometer (INCYTO, #DHC-N01). Low-passage (p<3) *B. burgdorferi* B31-A3 was used throughout this study.

## In vivo infection of mice

The *Slpi-/-* C57BL/6 mice have been described previously (*Matsuba et al., 2017*; *Nakamura et al., 2003*). The wild-type (WT) C57BL/6 mice were purchased from the Jackson Laboratory and used as the controls. 5–7-week-old female WT and *Slpi-/-* C57BL/6 mice were used for infection. 4–6-week-old female C3H/HeN mice were purchased from Charles River Laboratories and used for infection. Both C57BL/6 and C3H/HeN mice were infected with low-passage $10^5$ *B. burgdorferi* subcutaneously (5–9 mice/group). PBS sham-infected mice were used as controls. Mice were euthanized approximately 3 weeks post infection within a 3-day window (between 21 and 24 dpi) based on the feasibility and logistics of the laboratory. Ear punch biopsies were taken at 7, 14, and between 21–24 dpi to determine the infection burden in the skin. Between 21 and 24 dpi, mice were euthanized, and heart and joint tissues were collected to quantify the spirochete burden. The protocol for the use of mice was reviewed and approved by the Yale Animal Care and Use Committee.

## Quantification of *Borrelia* burden

DNA was extracted from the heart, tibiotarsal joint, and ear punch samples using QIAGEN DNeasy Blood & Tissue Kit, QIAGEN. Quantitative PCR was performed using iQ-SYBR Green Supermix (Bio-Rad). For quantitative detection of *B. burgdorferi* burden within mouse tissue samples, q-PCR was performed with DNA using flagellin (*flaB*), a marker gene for *Borrelia* detection. The mouse *β-actin*

gene was used to normalize the amount of DNA in each sample. The nucleotide sequences of the primers used in specific PCR applications were described previously (*Gupta et al., 2020*).

## Joint histopathology analysis

Mice were euthanized by $CO_2$ asphyxiation, and one rear leg from each mouse was dissected, immersion-fixed in Bouin's solution (Sigma-Aldrich, #HT10132). Fixed tissues were embedded, sectioned, and stained with H&E by routine methods (Comparative Pathology Research Core in the Department of Comparative Medicine, Yale School of Medicine). Periarticular and joint inflammation was scored in a blinded fashion in a graded manner from 0 (negative), 1 (minimal), 2 (moderate), to 3 (severe).

## Flow cytometry to quantify infiltrating cells in joint tissues in mice

The WT and *Slpi-/-* C57BL/6 mice were infected with *B. burgdorferi* as described above. The mice were euthanized between 21 and 24 dpi. The ankle joints were cut out at around 0.7 cm proximal to the ankle joint. The portion distal to the midfoot was discarded, and the skin removed. The bone marrow cells were flushed out with RPMI 1640 (Gibco, #11875-093) using a 27-gauge needle. The bone marrow-depleted ankles were cut into 3–4-mm-sized tissue pieces and incubated with digestion media containing 2.4 mg/ml hyaluronidase (Sigma-Aldrich, #H3506), 1 mg/ml collagenase (Sigma-Aldrich, #C2139) in RPMI 1640 (Gibco, #11875-093) supplemented with 10% fetal bovine serum (FBS) for 1 h at 37°C with 5% $CO_2$. The digestion media containing the tissue pieces were passed through a 70 μm cell strainer (Thermo Scientific, #352350). The remaining tissue pieces were mashed using a 10 ml syringe plunger. The digestion media containing the isolated cells were neutralized with RPMI 1640 with 10% FBS (*Akitsu and Iwakura, 2016*). The red blood cells were lysed by ACK Lysing buffer (Gibco, #A1049201). The cells were rinsed and resuspended in FACS buffer and ready for staining for flow cytometry.

The cells were incubated with Fc receptor antibody TruStain FcX anti-mouse CD16/32; BioLegend, #101320, and antibodies including PerCP anti-mouse CD45 (BioLegend, #103130), BV711 anti-mouse Ly6G (BioLegend, #127643), PE anti-mouse CD11b (BioLegend, #101208), APC/CY7 anti-mouse CX3CR21 (BioLegend, #149047), FITC anti-mouse Ly6C (BioLegend, #128005), APC anti-mouse CD64 (BioLegend, #139305), and LIVE/DEAD fixable violet stain kit (Invitrogen, #L34955) on ice for 30 min. The samples were rinsed twice with FACS buffer and run through BD LSRII (BD Biosciences). The data was then analyzed using FlowJo (*Feng et al., 2023*).

## Gene expression evaluation by quantitative real-time PCR

Mice were euthanized between 21 and 24 dpi. The ankle joints were excised as described above, snap-frozen in liquid nitrogen, and stored at −80°C. The frozen tissue was pulverized in liquid nitrogen using a mortar and pestle (*Wilson et al., 2021*). The RNA was purified using Trizol (Invitrogen, #15596-018) following a published protocol (*Zheng and McAlinden, 2021*). cDNA was synthesized using the iScript cDNA Synthesis Kits (Bio-Rad, #1708891). qPCR was performed using iQ SYBR Green Supermix (Bio-Rad, #1725124). The relative expression of each target gene was normalized to the mouse *β-actin* gene. The target genes and corresponding primer sequences are shown in the Key Resources Table.

## Murine neutrophil elastase, cytokine, chemokine, and MMP profile

Blood samples from each group of mice was collected by cardiac puncture between 21 and 24 dpi, and sera were collected. The murine neutrophil elastase level was measured using the Mouse Neutrophil Elastase/ELA2 DuoSet ELISA (R&D Systems, #DY4517-05). Serum was sent for cytokine analysis by the Mouse Cytokine/Chemokine 32-Plex Discovery Assay Array (MD32) and the Mouse MMP 5-Plex Discovery Assay Array (MDMMP-S, P) performed by Eve Technologies. The cytokines and chemokines represented by MD32 are Eotaxin, G-CSF, GM-CSF, IFN-γ, IL-1α, IL-1β, IL-2, IL-3, IL-4, IL-5,IL-6, IL-7, IL-9, IL-10, IL-12 (p40), IL-12 (p70), IL-13, IL-15, IL-17A, IP-10, KC, LIF, LIX, MCP-1, M-CSF, MIG, MIP-1α, MIP-1β, MIP-2, RANTES, TNFα, and VEGF. The MMPs represented by MDMMP-S, P are MMP-2, MMP-3, MMP-8, proMMP-9, and MMP-12.

## Purification of recombinant murine SLPI

The murine *Slpi* cDNA ORF clone was purchased from GenScript (OMu22721). The coding sequence was subsequently cloned into pET22b(+) expression vector (Novagen) in frame with the pelB signal peptide using Gibson Assembly (*Maffia et al., 2007*). *E. coli* strain Rosetta-gami 2 (DE3) (Novagen, #71351) was transformed with the SLPI-pET22b+ and grown at 37°C with ampicillin (100 µg/ml), tetracycline (12.5 µg/ml), streptomycin (50 µg/ml), and chloramphenicol (34 µg/ml). Cells were induced with 1 mM IPTG (18°C, overnight), harvested, and lysed with BugBuster Protein Extraction Reagent (Novagen, #70921-3). Recombinant mSLPI was purified with a Ni-NTA resin column as described by the manufacturer (QIAGEN). To evaluate the activity of the purified rmSLPI, the trypsin inhibitory activity was assayed with the fluorescent substrate Mca-RPKPVE-Nval-WRK(Dnp)-NH2 Fluorogenic MMP Substrate (R&D Systems, #ES002) and the absorbance was monitored at 405 nm using a fluorescent plate reader (Tecan).

## Flow cytometry to validate *B. burgdorferi*-SLPI binding

Actively growing low-passage *B. burgdorferi* was cultured to a density of $10^6$–$10^7$ cells/ml and harvested at 10,000 × *g* for 10 min. Cells were rinsed twice with PBS and blocked in 1% BSA for 1 h at 4°C. The cells were pelleted, rinsed, resuspended, and incubated with 10 nM and 1 µM human SLPI (R&D Systems, #1274-PI-100) and murine SLPI (produced in lab as described above) at 4°C for 2 h. The binding was detected with goat anti-human or murine SLPI (R&D Systems, #AF1274 and AF1735) and Alexa Fluor 488 or Alexa Fluor 647 donkey anti-goat IgG (H+L) (Invitrogen, #A32814, A-21447). The samples were fixed with 2% PFA before running through BD LSRII Green (BD Biosciences). The data was then analyzed using FlowJo. The integrity of *B. burgdorferi* organism was confirmed by Hoechst 33324/propidium iodide double staining (*Figure 5—figure supplement 3*). The fixed *B. burgdorferi* sample was included as a positive control for the propidium iodide staining. Permeabilization was not performed during this protocol. Thus, the binding detected was on the bacterial outer surface.

## ELISA to validate *B. burgdorferi*-SLPI binding

*B. burgdorferi* was cultured to a density of $10^6$–$10^7$ cells/ml and harvested at 10,000 × *g* for 10 min. To make the *B. burgdorferi* lysate, cells were rinsed twice with PBS, pelleted, and lysed using BugBuster Protein Extraction Reagent (Novagen, #70921-3). Protein concentration in the lysate was measured by absorbance at 280 nm using the nanodrop (Fisher Scientific). For the protease assay, the *B. burgdorferi* lysate was incubated in the presence or absence of proteinase K (0.2 mg/ml, Thermo Scientific, #EO0491) for 10 min. In an immuno 96-well plate (MaxiSorp), wells were coated with 200 ng of *B. burgdorferi* lysate. Samples were blocked with 1% BSA followed by incubation with human SLPI at varying concentrations (1–1000 ng) for 1 h at room temperature. The binding was probed with goat anti-human SLPI (R&D Systems, #AF1274) and rabbit anti-goat IgG (whole molecule)-peroxidases antibody (Sigma-Aldrich, #A8919-2ML). KPL Sureblue TMB Microwell Peroxidase substrate, 1-component (Seracare, #5120-0077) was used. The reaction was stopped with KPL TMB stop solution (Seracare, #5150-0021), and absorbance was read at 450 nm.

## Immunofluorescence assay

Actively growing low-passage *B. burgdorferi* was cultured to a density of $10^6$–$10^7$ cells/ml, rinsed twice with PBS, and blocked with 1% BSA for 1 h at 4°C. *B. burgdorferi* was incubated with human or murine SLPI at 4°C for 2 h. The spirochetes were probed with goat anti-human or murine SLPI (R&D Systems, #AF1274 and AF1735) and Alexa Fluor 488 donkey anti-goat IgG (H+L) (Invitrogen, #A32814). *B. burgdorferi* were then stained with Hoechst 33342 (Invitrogen, #H1399). The samples were fixed with 2% PFA before imaged with Leica SP8. The integrity of *B. burgdorferi* organism is confirmed in *Figure 5—figure supplement 3*. Permeabilization was not performed during this protocol. Thus, the binding detected was on the bacterial outer surface.

## BacTiter Glo microbial cell viability assay to quantify *B. burgdorferi* viability

The BacTiter Glo microbial cell viability assay quantifies the ATP present in the microbial culture by measuring luminescence. The amount of ATP is proportional to the number of viable cells in culture (*Gupta et al., 2020*; *Arora et al., 2022*; *Schwendinger et al., 2013*). To test the borreliacidal activity

of human SLPI, $1 \times 10^5$ spirochetes were treated with 0–10 μM hSLPI (R&D Systems, #1274-PI-100) at 33°C for 48 h. The luminescence was measured using a fluorescence plate reader (Tecan). The percent viability was normalized to the control spirochetes culture without hSLPI treatment. To test the effect of hSLPI on the antibody-mediated *B. burgdorferi* killing, $1 \times 10^5$ spirochetes were pretreated with 0–5 μM hSLPI (R&D Systems, #1274-PI-100) at 33°C for 2 h. 20% mouse *B. burgdorferi* antisera were then added and incubated for 2 and 4 h. The mouse antisera were collected from *B. burgdorferi*-infected mice between 21 and 24 dpi. The luminescence was measured as described above. The percent viability was normalized to the control spirochetes culture without any treatment.

## Statistical analysis

The analysis of all data was performed using the nonparametric Mann–Whitney or ANOVA using Prism 10 software (GraphPad Software, Inc, San Diego, CA). A p-value of <0.05 was considered statistically significant.

## Acknowledgements

We are grateful to Dr. Narasimhan for her input and suggestions during experiment design, and to Dr. Ming-Jie Wu for his assistance in conducting experiments. We are grateful to Ms. Ming Li for her effort in preparing human sera samples. This work was supported by NIH grants AI165499 and AI138949, the Steven and Alexandra Cohen Foundation, and the Howard Hughes Medical Institute Emerging Pathogens Initiative.

## Additional information

### Funding

| Funder | Grant reference number | Author |
|---|---|---|
| National Institute of Allergy and Infectious Diseases | AI165499 | Erol Fikrig |
| National Institute of Allergy and Infectious Diseases | AI138949 | Erol Fikrig |
| Steven and Alexandra Cohen Foundation | | Qian Yu Xiaotian Tang Thomas Hart Erol Fikrig |
| Howard Hughes Medical Institute | Emerging Pathogens Initiative | Qian Yu Xiaotian Tang Thomas Hart Erol Fikrig |

The funders had no role in study design, data collection and interpretation, or the decision to submit the work for publication.

### Author contributions

Qian Yu, Conceptualization, Data curation, Formal analysis, Validation, Investigation, Visualization, Methodology, Writing – original draft, Writing – review and editing; Xiaotian Tang, Conceptualization, Data curation, Investigation, Methodology, Writing – review and editing; Thomas Hart, Conceptualization, Resources, Formal analysis, Investigation, Methodology; Robert Homer, Data curation, Visualization; Alexia A Belperron, Linda K Bockenstedt, Resources, Writing – original draft, Writing – review and editing; Aaron Ring, Akira Nakamura, Resources, Methodology; Erol Fikrig, Conceptualization, Resources, Data curation, Supervision, Funding acquisition, Investigation, Methodology, Writing – original draft, Project administration, Writing – review and editing

### Author ORCIDs

Qian Yu ⓘ https://orcid.org/0000-0002-2396-2128
Xiaotian Tang ⓘ https://orcid.org/0000-0002-0171-9354
Robert Homer ⓘ https://orcid.org/0000-0002-2055-5885

Linda K Bockenstedt ⓘ https://orcid.org/0000-0001-9349-8757

### Ethics

This study used archived serum samples from adult Lyme disease subjects and healthy controls that were previously collected under NIH U19AI089992 with approval of the Yale University Institutional Review Board for human subjects research (IRB protocol# 1112009475).

All the animal experiments in this study were performed in accordance with the Guidelines for the Care and Use of Laboratory Animals of the National Institutes of Health. The animal protocol (2023-07941) was approved by the Institutional Animal Care and Use Committee at the Yale University School of Medicine.

Reviewer #2 (Public review): https://doi.org/10.7554/eLife.104913.4.sa1

Author response https://doi.org/10.7554/eLife.104913.4.sa2

## Additional files

### Supplementary files

Supplementary file 1. Subject characterization.

MDAR checklist

### Data availability

All data generated or analyzed in this study are included in the manuscript and the supplementary files.

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
