## [Editor Report · eLife Assessment]

This study presents a **valuable** finding on the role of secretory leukocyte protease inhibitors (SLPIs) in developing Lyme disease in mice infected with *Borrelia burgdorferi*. The evidence supporting the authors' claims is **solid**. However, several concerns raised by the reviewers remain unaddressed. This paper would be of interest to scientists in the infectious inflammatory disease field.

---

## [Referee Report · Reviewer #2 (Public review)]

This study by Yu and coworkers investigates the potential role of Secretory leukocyte protease inhibitor (SLPI) in Lyme arthritis. They show that, after needle inoculation of the Lyme disease agent, B. burgdorferi, compared to wild type mice, a SLPI-deficient mouse suffers elevated bacterial burden, joint swelling and inflammation, pro-inflammatory cytokines in the joint, and levels of serum neutrophil elastase (NE). They suggest that SLPI levels of Lyme disease patients are diminished relative to healthy controls. Finally, using a powerful screen of secreted mammalian proteins, they find that SLPI interacts directly B. burgdorferi.

The known role of SLPI in dampening inflammation and inflammatory damage by inhibition of NE makes the enhanced inflammation in the joint of B. burgdorferi-infected mice a predicted result but it has not previously been demonstrated and could spur further study. A limitation that is unaddressed experimentally is potential contribution of the greater bacterial burden to the enhanced inflammation, leaving open the question of whether greater immunologic stimulus or a defect in the regulation of inflammation is responsible for the observed enhanced disease. Answering this question would better justify the statement in the abstract that "These data demonstrate the importance of SLPI in suppressing periarticular joint inflammation in Lyme disease."

Although the finding of SLPI binding to bacteria is potentially quite interesting the biological relevance of this interaction is not addressed. Readers of only the abstract, which describes the direct interaction of SLPI with bacteria, may mistakenly conclude that the authors demonstrate that recruitment of this immunoregulatory factor to the bacterial surface enhances inflammation of infected tissues. This attractive possibility has not been demonstrated in this study; such assertion would require comparison of bacteria that either bind or do not bind SLPI in a mouse infection model.

Finally, the investigators take advantage of clinical samples to ask if serum SLPI levels a diminished in Lyme disease patients relative to healthy controls. The assessment of human samples is interesting and generally to be lauded, but here the comparison is limited by: (a) a small sample number, with only 5 healthy control samples (which should not be difficult to obtain); and (b) the inclusion of samples from 4 patients with erythema migrans rather than Lyme arthritis, which was the manifestation tracked in the mouse studies. Moreover, of the 3 Lyme arthritis patients, serum samples from multiple blood draws were included, resulting in 5 data points; similarly, of the 4 erythema migrans patients, 13 separate samples were included. The multiple samplings from some but not all subjects could result in differential "weighting" of samples. Therefore, although the investigators provide a statistical analysis of these data, it is difficult to evaluate the validity of this apparent difference.

In summary, this is an interesting study that provides new information regarding infection in a host deficient in SLPI and, using a state-of-the-art screen of the mammalian secretome to show that B. burgdorferi binds SLPI, raising the attractive possibility that this pathogen utilizes a host immune regulator to enhance inflammation. The conclusions that SLPI enhances inflammation directly due to its immunoregulatory activity and that SLPI levels are diminished in human Lyme disease patients, as well as the implication that SLPI binding by the bacterium has pathogenic significance, each require further study.

---

## [Author Response]

The following is the authors’ response to the current reviews.

We deeply appreciate the reviewer’s careful review and critiques. These are excellent critiques that we are working on and probably require a few more years of work. Published together, we believe these critiques add value to our manuscript.

The following is the authors’ response to the original reviews.

**Reviewer #2 (Public review):**
Summary:This manuscript by Yu and coworkers investigates the potential role of Secretory leukocyte protease inhibitor (SLPI) in Lyme arthritis. They show that, after needle inoculation of the Lyme disease (LD) agent, B. burgdorferi, compared to wild type mice, a SLPI-deficient mouse suffers elevated bacterial burden, joint swelling and inflammation, pro-inflammatory cytokines in the joint, and levels of serum neutrophil elastase (NE). They suggest that SLPI levels of Lyme disease patients are diminished relative to healthy controls. Finally, they find that SLPI may interact directly the B. burgdorferi.Strengths:Many of these observations are interesting and the use of SLPI-deficient mice is useful (and has not previously been done).Weaknesses:(a) The known role of SLPI in dampening inflammation and inflammatory damage by inhibition of NE makes the enhanced inflammation in the joint of B. burgdorferi-infected mice a predicted result; (b) The potential contribution of the greater bacterial burden to the enhanced inflammation is acknowledged but not experimentally addressed; (c) The relationship of SLPI binding by B. burgdorferi to the enhanced disease of SLPI-deficient mice is not addressed in this study, making the inclusion of this observation in this manuscript incomplete; and (d) assessment of SLPI levels in healthy controls vs. Lyme disease patients is inadequate.

We greatly appreciate the critiques, and we do agree. Even though the observation of NE level is predictable, we believe that it is important to actually demonstrate it in the context of murine Lyme arthritis. The function of SLPI goes beyond inhibiting NE level. As an ongoing project in our lab, we believe that the current study serves as a good starting point to explore the pleiotropic effects SLPI in the pathogenesis of murine Lyme arthritis and in patients. And, the critiques here are of great value to our research.

Comments on revised version:Several of the points were addressed in the revised manuscript, but the following issues remain:Previous point that the relationship of SLPI binding to B. burgdorferi to the enhanced disease of SLPI-deficient mice is not investigated: The authors indicate that such investigations are ongoing. In the absence of any findings, I recommend that their interesting BASEHIT and subsequent studies be presented in a future study, which would have high impact.

We thank the reviewer for the critique. We do agree that this part of the story is not complete. However, we would like to keep the BASEHIT and binding data in the paper, as we believe that it is an important finding. We confirmed the binding using ELISA, flow cytometry, and immunofluorescent microscopy. We showed that the binding is specific to infectious strain of B. burgdorferi, thus likely to contribute to the pathogenesis of murine Lyme arthritis. Our data suggest that SLPI can directly interact with a B. burgdorferi protein. We are exploring the biological significance of the binding. And this finding can be further explored by other labs too.

Previous recommendation 1: (The authors added lines 267-68, not 287-68). This ambiguity is acknowledged but remains. In addition, in the revised manuscript, the authors state "However, these data also emphasize the importance of SLPI in controlling the development of inflammation in periarticular tissues of B. burgdorferi-infected mice." Given acknowledged limitations of interpretation, "suggest" would be more appropriate than "emphasize".

We thank the reviewer for the careful reading, and we apologize for the mistake. The change has been made accordingly (line 268).

Previous recommendation 5: The lack of clinical samples can be a challenge. Nevertheless, 4 of the 7 samples from LD patients are from individuals suffering from EM rather than arthritis (i.e., the manifestation that is the topic of the study) and some who are sampled multiple times, make an objective statistical comparison difficult. I don't have a suggestion as to how to address the difference in number of samples from a given subject. However, the authors could consider segregating EM vs. LA in their analysis (although it appears that limiting the comparison between HC and LA patients would not reveal a statistical difference).

We thank the reviewer for the critique. And we agree with the reviewer that the patient’s data presented are not ideal. We believe that at this point the combination of the samples is most logical, as the number of samples we have from patients with Lyme arthritis is fairly limited. We stated the limitation in the discussion. We do believe that the finding of the correlation is important. It suggests the potential function of SLPI in patients, beyond murine infection.

What’s more, various groups with large number of different samples can elucidate the relationship further.

Previous recommendation 6: Given that binding of SLPI to the bacterial surface is an essential aspect of the authors' model, and that the ELISA assay to indicate SLPI binding used cell lysates rather than intact bacteria, a control PI staining to validate the integrity of bacteria seems reasonable.

We appreciate the suggestion and has provided the propidium iodide staining in Supplemental Figure 5 (line 539-542, 568-569, 718-722).

Previous recommendation 8: The inclusion of a no serum control (that presumably shows 100% viability) would validate the authors' assertion that 20% serum has bactericidal activity.

We appreciate the suggestion. As stated in the manuscript (line 583-584), the percent viability was normalized to the control spirochetes culture without any treatment. Thus, the control spirochetes culture, without serum and SLPI treatment, showed 100% viability. We have revised Supplemental Figure 3 accordingly.